# Transfer Learning via Context-aware Feature Compensation

## Abstract

Transfer learning aims to reuse the learnt representations or subnetworks to a new domain with minimum effort for adaption. Here, the challenge lies in the mismatch between source domain and target domain, which is the major gap to be tackled by transfer learning. Hence, how to identify the mismatch between source and target domain becomes a critical problem. We propose an end-to-end framework to learn feature compensation for transfer learning with soft gating to decide whether and how much feature compensation is needed, accounting for the mismatch between source domain and target domain. To enable identifying the position of the input in reference to the overall data distribution of source domain, we perform clustering at first to figure out the data distribution in a compact form represented by cluster centers, and then use the similarities between the input and the cluster centers to describe the relative position of the input in reference to the cluster centers. This acts as the context to indicate whether and how much feature compensation is needed for the input to compensate for the mismatch between source domain and target domain. To approach that, we add only two subnetworks in the form of Multilayer Perceptron, one for computing the feature compensation and the other for soft gating the compensation, where both are computed based on the context. The experiments show that such minor change to backbone network can result in significant performance improvements compared with the baselines on some widely used benchmarks.

## 1 Introduction

Transfer learning aims to reuse the knowledge obtained from one domain to solve the problem in a new domain with little effort and minor change. A classical problem that needs intensive effort is pattern recognition, where training of a model consumes a lot of time and requires a large number of data examples with annotations. If we can reuse the pre-trained model for another task, it leads to efficient and rapid development of a new solution. For example, a speech recognition model trained on one language can be transferred to recognize another language more easily by using transfer learning (Huang et al., 2013). In addition, the knowledge learnt for visual navigation can be transferred to a new environment with ease through transfer learning (Al-Halah et al., 2022). So far, a lot of transfer learning methods have been proposed and it has been attracting much attention.

In general, a pattern recognition system is composed of feature extraction and classifier. In the context of deep neural networks with end-to-end learning, some portions of a network act as filters to perform feature extraction while the last layer is in general a fully-connected layer to conduction classification. One solution for transfer learning is to share the feature extraction portion of the network and apply different classifiers for different tasks (Oquab et al., 2014). When there is a big gap between the source and target domain in terms of feature distribution, simply modifying classifier will fail to solve the mismatch between the two domains. Hence, another solution aims to identify reusable features or subnetworks performing feature extraction (Huang et al., 2013). Yet, this relies on the assumption that there exist some coherent features that can be shared by different domains. In view of the distinction between source and target domain in terms of feature representation, another solution tries to make the features of two domains approach each other through representation learning, for example, applying certain regularization to loss function (Zhong & Maki, 2020) or a domain classifier (Ajakan et al., 2014) to perform adversarial test on homogeneous or not. Even though, they are still based on such assumption that feature distributions of two domains can have

substantial overlap. However, this hypothesis might not always hold significantly in practice. Here-after, another solution to modify the network structure arises, for example, using an agent to search for reusable subnetworks to form a new pipeline for target domain, where the agent could be implemented using neural networks (Guo et al., 2020; Liu et al., 2021). However, these works change the backbone network greatly, which deviates from the origin of transfer learning, that is, solve new problem with off-the-shelf solution by paying minimum effort to make tiny change.

As aforementioned, the critical issue in terms of transfer learning is to identify the reusable part of the existing learnt representations or subnetworks. In another word, due to the overlap between source domain and target domain, not every data example needs transferring, and how much transferring is needed for a data example is subject to its position in the overall data distribution. If the data example of interest locates in the overlap part between source domain and target domain, it needs minor change. On the contrary, if the data example exists in a feature space where the data distribution of source domain is significantly different that of target domain, it needs significant change through transfer learning correspondingly. This gives rise to a new problem, that is, the method to evaluate how much mismatch exists around the input between source domain and target domain. This involves development of a means to figure out the overall data distribution as well as a descriptor to locate the position of the input in reference to the overall data distribution, say, context. With sound context description in terms of figuring out the position of the input in regard to the overall data distribution, one can then proceed to evaluate whether and how much transferring is needed to compensate for the mismatch between source domain and target domain. For this sake, we propose a new method for context description: We apply clustering at first to represent the data distribution in a compact form of cluster centers, and then compute the similarities between the input and the cluster centers to define its location in the overall data distribution, which acts as a clue to lead subsequent computation on how much compensation is needed to enhance the feature representation from backbone network. Based on such a scheme for context description, we propose an end-to-end framework to learn feature compensation for transfer learning with soft gating to decide whether and how much feature compensation is needed, accounting for the mismatch between source domain and target domain. To approach that, we add only two subnetworks in the form of Multilayer Perceptron (MLP) to backbone network, one for computing the feature compensation and the other for soft gating the compensation, where both are computed based on the context, that is, the feature compensation is context adaptive.

The contribution of this study is as follows:

1. A novel method is proposed for context-aware feature compensation, where similarities between the input and the cluster centers representing the overall data distribution is used as the clue to evaluate how much feature compensation is needed to solve the mismatch between source domain and target domain. The new architecture only incorporates two additional MLP modules, one for computing feature compensation and the other for soft gating the compensation. This meets almost all the requirements of transfer learning: Minor network-level change with two tiny components added, feature-level compensation with context awareness on whether and how much transferring is needed, and minimum effort to render a sound solution with big performance improvement;

2. The concept of context based on self-positioning in reference to the anchors (cluster centers) is proposed for figuring out the demanding degree in transferring the representation of each given example. Correspondingly, feature compensation is computed by incorporating the context into the existing representation while a soft gating is applied to highlight the degree that each data example needs transferring, where both the context-directed feature compensation and the soft gating are leant end to end;

3. The experiments on 5 data sets demonstrate the effectiveness as well as the advantages of our solution in comparison with the baselines.

## 2 RELATED WORK

**Transfer learning** aims to transfer the learnt knowledge from source domain to target domain (Thrun, 1995) so as to allow pre-trained models reused with minimum effort and good transferability. The existing methods can be sorted into 4 categories (Tan et al., 2018): Instances-based, mapping-based, adversarial-based, and network-based deep transfer learning.

Instances-based deep transfer learning aims to reuse the instances of source domain. For example, TrAdaBoost (Dai Wenyuan et al., 2007) weights the instances of source domain to make the distribution similar to that of target domain, and perform model training based on the weighted instances from source domain along with those from target domain.

Mapping-based deep transfer learning maps the source domain and target domain to a new space, in which they could be coherent to each other to some extent, and model is trained in the new space. A representative work in this direction is Transfer Component Analysis (Pan et al., 2010), and Tzeng et al. (Tzeng et al., 2014) extents it to deep learning based solutions.

Adversarial-based deep transfer learning is based on Generative Adversarial Network (GAN) (Goodfellow et al., 2014) to search the transferring solution from source to target domain. Domain Adversarial Neural Network (DANN) (Ajakan et al., 2014) introduces the idea of GAN to transfer learning by applying a domain classifier to minimize the difference between source and target domain in feature learning, that is, optimization on transferred feature representation is driven by not only classification but also removing the gap between source and target domain.

Network-based deep transfer learning reuses the model pre-trained on source domain, including architecture and parameters, to form a part of the solution of target domain, since filters in convolutional layers act as feature extractors in fact and the feature embedding turned out from them are regarded generic to some extent. An early work for speech recognition (Huang et al., 2013) partitions the neural networks of a solution into two parts: Language-irrelevant and language relevant, and reuse the language-irrelevant part for other recognition tasks. Oquab et al. (Oquab et al., 2014) reuses the Convolutional Neural Network (CNN) pre-trained on ImageNet with additional fully connected layers to adapt to target-domain classification. Another classical method, namely fine tune (Yosinski et al., 2014), reuses the original network without any change on the structure and just modifies the parameter values with the data from target domain, where fine tune can be performed on the whole network, or a part of them after fixing the other part. There are two tracks to further improve (Yosinski et al., 2014): Some methods (Xuhong et al., 2018; Li et al., 2019; Zhu et al., 2018; Zhong & Maki, 2020) advance it by introducing new loss functions and the others (Guo et al., 2020; Liu et al., 2021) achieve advance by optimizing the network structure. Fine tune is limited in that the difference between source domain and target domain is not taken into account. In view of that, $L^2$-SP (Xuhong et al., 2018) and DELTA (Li et al., 2019) apply regularization to loss function in order to compensate for such difference. GTN (Zhu et al., 2018) holds in terms of feature selection to weight features with gating neural network such that the common features shared by source domain and target domain can receive more attention. PtR (Zhong & Maki, 2020) aims to optimize feature learning by altering loss function to include regularization on the fly when the training is approaching a sound solution. Head2Toe (Evci et al., 2022) makes use of the representations from all layers and applies Lasso to obtain feature selection in their transfer learning framework. From another point of view, parameter modification on the original network (Xuhong et al., 2018; Li et al., 2019; Zhu et al., 2018; Zhong & Maki, 2020) is not able to tackle all mismatches between source and target domain. Hereafter, AdaFilter (Guo et al., 2020) and TransTailor (Liu et al., 2021) alleviate this by reorganizing the network architecture. As confirmed in (Iofinova et al., 2022), pruning the network into sparse ones in general leads to better discriminant for downstream tasks. AdaFilter applies Layer-wise Recurrent Gated Network to select reusable filters in original solution to form a new pipeline as target-domain solution. TransTailor introduces pruning to make network structure as concise as enough for target domain, which leads to a optimal sub-model. However, they change the network substantially, which deviates a bit from the origin of transfer learning, that is, gain better discriminant with minor change and little effort. In contrast to (Guo et al., 2020; Liu et al., 2021), our method can approach this goal since we just add a tiny network to obtain enhanced feature to compensate for the feature from backbone network. Compared with the Pseudo label used in (Zhong & Maki, 2020), we use the similarity between the data example of interest to all the centers from clustering, which figures out the contextual situation regarding feature discriminant more comprehensively than a simple Pseudo label.

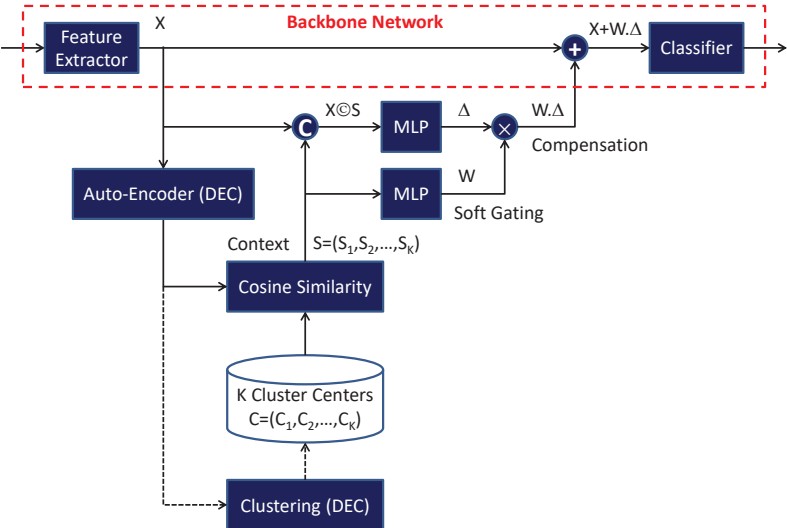

MLP: Multilayer Perceptron; ©: Concatenation; ⊗: Scalar Product; ⊕: Add; ---→ : Training Only

Figure 1: Context-directed Transfer Learning

## 3 METHOD

### 3.1 FRAMEWORK OF TRANSFER LEARNING

Our motivation is simple: For a given example, whether and how much the corresponding embedding should be changed to approach a new task is subject to its position in the data distribution. If its position is closer to an existing cluster center, in general, a higher probability of correct classification can be expected due to the explicit association with known data representatives. In such a case, the embedding should be changed little. On the contrary, the embedding should be remedied remarkably if its association to any cluster center is not so deterministic. Therefore, we first perform clustering on training data to obtain the centers representing data distribution. Then, we compute the similarity between the input data example and each of the centroids resulting from clustering so as to figure out the position of the input in reference to the cluster centers, where the cluster centers act as anchors to figure out data distribution as well as how far from the input to each anchor. We denote the $K$ cluster centroids as $\boldsymbol{C} = (\boldsymbol{C}_1, \boldsymbol{C}_2, \ldots, \boldsymbol{C}_K)$ and the input as $\boldsymbol{X}$. Then, we compute the cosine similarity between $\boldsymbol{X}$ and $\boldsymbol{C}_1, \boldsymbol{C}_2, \ldots, \boldsymbol{C}_K$ respectively, and the corresponding similarity is denoted as $\boldsymbol{S} = (\boldsymbol{S}_1, \boldsymbol{S}_2, \ldots, \boldsymbol{S}_K)$. Note that $\boldsymbol{X}$ should undergo Auto-Encoder based representation learning prior to clustering and similarity computing, as required by the clustering module applied in this study, namely, Deep Embedded Clustering (DEC) (Xie et al., 2016). These similarities are referred to as context since they figure out the relative position of $\boldsymbol{X}$ in regard to the cluster centers, based on which the compensation to the original feature embedding is computed. As shown in Figure 1, the compensation to the original embedding is subject to two branches: The first branch appends the context vector $\boldsymbol{S}$ to the original embedding $\boldsymbol{X}$ via concatenation, and then a multilayer perceptron (MLP) is used to transform $\boldsymbol{X} © \boldsymbol{S}$ into $\Delta = MLP(\boldsymbol{X} © \boldsymbol{S})$, where MLP is designed to force $\Delta$ has the same dimension as $\boldsymbol{X}$, and © denotes concatenation. The second branch computes the soft gating factor $\boldsymbol{W} = MLP(\boldsymbol{S})$ so as to obtain the weighted compensation subject to the context, that is, $\boldsymbol{W}.\Delta$, where $\boldsymbol{W}$ is a scalar. Following the aforementioned transfer network, the overall embedding fed to classifier becomes:

$$\boldsymbol{X} + \boldsymbol{W}.\Delta = \boldsymbol{X} + MLP(\boldsymbol{S}).MLP(\boldsymbol{X} © \boldsymbol{S}) \tag{1}$$

The transfer network to produce $\boldsymbol{W}.\Delta = MLP(\boldsymbol{S}).MLP(\boldsymbol{X} © \boldsymbol{S})$ is trained end to end. Once $W = 0$, the overall embedding in Eq. (1) degrades to that of backbone network, and a higher $W$ corresponds with a bigger transfer in terms of compensating the embedding. After applying transfer learning, the classification changes from $\boldsymbol{y} = \phi(\boldsymbol{X})$ to $\boldsymbol{y} = \phi(\boldsymbol{X} + \boldsymbol{W}.\Delta)$, where $\phi$ denotes classifier and $\boldsymbol{y}$ the class label resulting from the classifier.

### 3.2 Anchors for Context Description

In Eq. (1), we concatenate $X$ and $S$ to produce the compensation $MLP(X \copyright S)$ for transferring feature embedding because $S$ indicates how ambiguous $X$ is in terms of being associate with a deterministic cluster, say, context. Moreover, the soft weighting $MLP(S)$ aims to control the overall compensation adaptive to the context $S$. It is apparent that anchors resulting from clustering determines the performance of the proposed solution since they are the base to figure out data distribution and aware of the contextual clue to produce compensation adaptively. Here, we make use of the algorithm referred to as Deep Embedded Clustering (DEC) (Xie et al., 2016) to obtain such anchor points, and the experiments confirm that DEC performs very well in the proposed framework.

### 3.3 Working Pipelines

The proposed transfer learning framework can work in 4 manners to form the following solutions:

- (I) At first, fine tune the backbone network, and then fix the backbone and train the transfer network only.
- (II) Train the backbone and transfer network as a whole end to end.
- (III) Fix the backbone network and just train the transfer network.
- (I+) Except for the clustering, which is performed on the whole training data without random sampling for Caltech 256 data, the other implementation is the same as (I)

## 4 Experiments

### 4.1 Data Sets

We use the following 5 data sets for performance evaluation as they are widely used for evaluating transfer learning algorithms, especially the baselines peered in this study sunch as TransTailor (Liu et al., 2021):

- Caltech 256-30 & Caltech 256-60 (Griffin et al., 2007) data sets contain 30,607 images of 256 objects. Due to the unbalanced data distribution for each category, the data are sampled at random to form a balanced data set for training with 30/60 data examples per class (Xuhong et al., 2018; Li et al., 2019), referred to as Caltech 256-30/Caltech 256-60.
- CUB-200 (Wah et al., 2011) data set contains 11,788 images of 200 categories of birds.
- MIT Indoor-67 (Quattoni & Torralba, 2009) data set includes 67 indoor scenes with 80 training examples and 20 testing data examples per class.
- Stanford Dogs (Khosla et al., 2011) data set contains 20,580 images of 120 categories of dogs, where 12,000 are used for training and the rest 8,580 for testing.

### 4.2 Backbone

ResNet-50 (He et al., 2016), ResNet-101 (He et al., 2016), and VGG-16 (Simonyan & Zisserman, 2014) pre-trained on ImageNet-1k (Deng et al., 2009) are used as backbone network, respectively.

### 4.3 Baselines

- TransTailor (Liu et al., 2021): Branch pruning is performed iteratively to remove filters of less importance in order to approach an optimal sub-model.
- AdaFilter (Guo et al., 2020): Layer-wise Recurrent Gated Network is used to select the pre-trained as well as fine-tuned filters reusable to form a new pipeline.
- GTN (Zhu et al., 2018): Gated Transfer Network is used to weight features from pre-trained model, which works in the sense of feature selection.
- PtR (Zhong & Maki, 2020): Pseudo-regression Task Loss is combined with Cross Entropy Loss as a regularization term once the training approaches a sound solution, by which

Table 1: Classification precision (%) compared with baselines on 5 data sets

| Backbone | Method | Caltech 256-30 | Caltech 256-60 | CUB-200 | Stanford Dogs | MIT Indoor-67 |
|----------|--------|----------------|----------------|---------|---------------|---------------|
| ResNet101 | TransTailor | 85.3 | 87.3 | 80.7 | 91 | 78.2 |
| | PtR | 84.5 | 87.2 | - | - | 79.2 |
| | $L^2$ | 81.5±0.2 | 85.3±0.2 | - | 81.4±0.2 | 79.6±0.5* |
| | $L^2$-SP | 83.5±0.1 | 86.4±0.2 | - | 85.1±0.2 | 84.2±0.3* |
| | DELTA | 86.6 | **88.7** | 80.5 | 88.7 | 85.5* |
| | Fine tune | 82.65 | 85.91 | 83.7 | 91.17 | **80.82** |
| | Ours (I) | 83.43 | 86.36 | 83.91 | **92.24** | 80.22 |
| | Ours (II) | 83.02 | 85.27 | **84.89** | 91.75 | 80.52 |
| | Ours (I+) | **86.62** | 86.66 | - | - | - |
| ResNet50 | PtR | **83.9** | **87.1** | 81.9 | - | 77.9 |
| | AdaFilter | 80.62 | 84.31 | - | 82.44 | 77.53 |
| | GTN | - | - | 79.95 | - | 77.77 |
| | $L^2$-SP | - | - | 78.44 | 86.72 | - |
| | DELTA | - | - | 78.63 | 86.01 | - |
| | BSS | - | - | 78.85 | 87.18 | - |
| | StochNorm | - | - | 79.58 | - | - |
| | Co-Tuning | - | - | 81.24 | - | - |
| | Fine tune | 79.94 | 83.04 | 83.83 | 89.79 | **79.55** |
| | Ours (I) | 80.58 | 83.39 | **83.98** | 90.5 | 79.4 |
| | Ours (II) | 81.09 | 83.73 | 83.93 | **90.92** | 79.62 |
| | Ours (I+) | 83.32 | 83.96 | - | - | - |
| VGG16 | TransTailor | **76.4** | **81.8** | 79.2 | **84.2** | **76.5** |
| | PtR | - | - | 75.1 | - | 73 |
| | GTN | - | - | 72.14 | - | 71.19 |
| | Fine tune | 54.06 | 59.14 | 76.57 | 75.5 | 72.91 |
| | Ours (I) | 52.61 | 58.47 | 76.51 | 75.81 | 72.23 |
| | Ours (II) | 60.46 | 66.42 | **80.3** | 77.65 | 75.59 |
| | Ours (III) | 66.46 | 70.66 | 64.84 | 83.39 | 71.34 |
| | Ours (I+) | 63.37 | 66.28 | - | - | - |

feature learning takes into account not only the classification perspective but also the transferability of features.

- $L^2$-SP (Xuhong et al., 2018): $L^2$–SP penalty is applied to the parameters of a pre-trained model in the form of regularization so as to enforce the weights hold sparsely.

- DELTA (Li et al., 2019): DELTA characterizes the distance between source/target networks using their outer layer outputs, and incorporates such distance as the regularization term of the loss function.

- BSS (Chen et al., 2019): Batch Spectral Shrinkage (BSS) inhibits negative transfer by suppressing the spectral components with small singular values that correspond to detrimental pre-trained knowledge.

- StochNorm (Kou et al., 2020): Stochastic Normalization (StochNorm) uses a two-branch normalization architecture to avoid model over-depending on biased statistics.

- Co-Tuning (You et al., 2020): The pre-trained model is fine-tuned using ground truth labels and probabilistic labels with learned relationship between source categories and target categories.

## 4.4 EXPERIMENTAL SETTING

All the experiments are conducted using PyTorch, running on NVIDIA GeForce RTX™ 3090 GPU for training. The pre-trained models of ResNet and VGG are provided by Timm (Wightman, 2019). Every image is resized to 224×224 resolution with every color channel normalized to zero mean. Training is based on SGD (stochastic gradient descent) with decaying rate being 0.005, momentum being 0.9, and learning rate being 0.0003. For our solution, the number of cluster centers is set to the true class number of the corresponding data set, which is 256, 200, 120, and 67 for Caltech 256, CUB-200, Stanford Dogs, and MIT Indoor-67, respectively.

## 4.5 PERFORMANCE COMPARISON

We list in Table 1 the performance comparison with baselines. Here, we compare these methods from two perspectives: (1) We compare different methods with a given backbone network: ResNet-101, ResNet-50, and VGG-16, respectively; (2) We compare the overall performance across all backbone models.

**ResNet-101:** On CUB-200 and Stanford dog data, the proposed method outperforms the baselines. For CUB-200 data, the precision improvement is about 4% compared with TransTailor, which is significant.

For Caltech 256-30 data, only 30 data samples are randomly selected per class for training. This degrades our method much because we use clustering to select representatives but the random selection prevents our method from finding the truly representative examples due to the too sparse random sampling of 30 data examples per class in contrast to the 30,607 images in total. To verify this issue, we reestablish our solution (I) by training the clustering model with all the data samples other than the testing samples, which is referred to as (I+) in Table 1. Note that the backbone network is stilly trained using the randomly sample 30 data examples per class. Consequently, the performance goes up to 86.62%, which outperforms all the baselines.

For Caltech 256-60 data, DELTA performs best. For our solution (I+), however, its performance improvement compared with that on Caltech 256-30 is minor as the full training data based clustering has lead to the best extent to figure out the data distribution, and the minor improvement comes from the backbone trained with more training data.

For MIT Indoor-67 data, the target is scene classification, which differs much from the source domain, say, object recognition. This might account for why the other methods fail to outperform fine tune in this case. Note that we annotate $L^2$, $L^2$-SP and DELTA with * in Table 1 as they use Places 365 (Zhou et al., 2017) as source-domain data for pre-training, which is also a scene classification data. As a result, their performance seems more promising since the source domain and the target domain are focused on the same visual recognition task.

**ResNet-50:** The result as well as the conclusion is similar to that of ResNet-101 based test in regard to our solution, and the exception is: PtR performs best on Caltech 256-30/60.

**VGG-16:** Our solution outperforms all baselines only on CUB-200 data, and is close to that of TransTailor on Stanford Dogs Data. For the other cases, the precision our model is quite lower compared with the baselines. However, VGG-16 is not a state-of-the-art model, not easy to train, and its performance is remarkably inferior to that of ResNet-101 according to Table 1. We see that all the best performances are with ResNet-101.

**Overall Comparison:** Regarding all backbones, ResNet-101 leads to the best performance on each data set. The result listed in Table 1 shows that our model performs best and achieves remarkable performance improvement on Caltech 256-30, CUB-200, and Stanford Dogs Data. For Caltech 256-60 and MIT Indoor-67, the performance of our solution is acceptable. In an overall sense, our solution is technically sound and performs better in comparison with the state-of-the-art methods.

Table 2: Classification precision (%) under different settings

| W= | Clustering | △= | Caltech 256-30 | Caltech 256-60 | CUB-200 | Stanford Dogs | MIT Indoor-67 |
|---|---|---|---|---|---|---|---|
| MLP(S) | DEC | MLP(X©S) | 83.02 | 85.27 | 84.89 | **91.75** | 80.52 |
| 1 | DEC | MLP(X©S) | 82.96 | 85.44 | 84.6 | 91.5 | **81.11** |
| MLP(S) | Random | MLP(X©S) | 83.49 | **86.05** | 84.53 | 91.41 | 80.89 |
| MLP(S) | K-Means | MLP(X©S) | **84.04** | 85.93 | **85.07** | 91.39 | 80.29 |
| MLP(S) | DEC | MLP(X) | 83.24 | 85.78 | 84.58 | 91.63 | 80.82 |

## 4.6 ABLATION STUDY

We conduct ablation study as follows: According to Table 2, the full model (solution II) outperforms the other settings on CUB-200 and Stanford Dogs data. For MIT Indoor-67 data, it seems that the

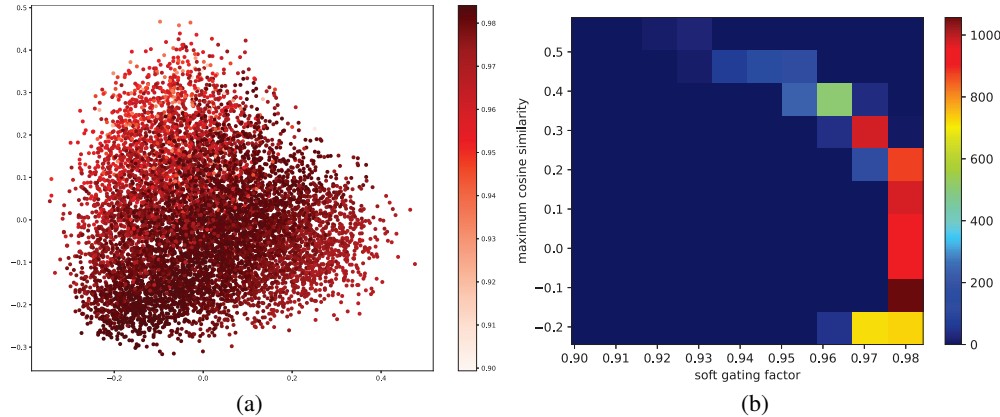

(a)  (b)

Figure 2: (a) Visualization of the data distribution of Stanford Dogs data via PCA. The heat value of each pixel corresponds with the value of the soft gating factor $W$ in association with this pixel. (b) 2-dimensional histogram of maximum cosine similarity against soft gating factor ($W$).

embedding of every data sample should be transferred, where W=1 leads to the best performance. Since this data set is developed for scene classification with layout not fixed between objects, this causes the target domain totally different from the source domain. In such a case, every data example undergoing transfer learning is sound. For the Caltech 256 data, 30/60 data samples per class are randomly selected for training. Since so few data are too sparse to figure out fully the actual distribution of the data, it is impossible for any clustering algorithm to capture the truly representative centers. In such cases, randomly selected representatives even work better for Caltech 256-60 data as shown in Table 2. For Caltech 256-30 and CUB-200 data, clustering with K-Means leads to the best performance while DEC performs best on the other two data sets. This means that clustering works in our framework but which one is more suitable merits further studies. Yet, DEC fits the end-to-end learning well due to its neural network nature.

Table 3: Classification precision against number of cluster centers on CUB-200 data

| #Centers | 200 | 300 | 400 | 500 | 600 | 700 |
|---|---|---|---|---|---|---|
| % | **84.89** | 83.98 | 84.69 | 84.48 | 84.74 | 84.08 |

Table 4: Classification precision against feature dimensionality arising from DEC Auto-Encoder on CUB-200 data

| Dim | 100 | 200 | 300 | 400 | 500 | 600 | 700 |
|---|---|---|---|---|---|---|---|
| % | **84.89** | 84.32 | 84.27 | 84.19 | 84.24 | 84.41 | 84.79 |

## 4.7 SENSITIVITY

As shown in Table 3, the number of cluster centers does not affect the performance much and the best one is 200. This means that the contextual description is robust, not subject to the number of cluster centers. Further, Table 4 shows that the performance change is minor with different feature dimensionality of DEC Auto-Encoder.

## 4.8 VISUALIZATION

We illustrated in Figure 2(a) the distribution of Stanford Dogs data, where the dimensionality of the feature space is reduced via Principal Component Analysis (PCA) and only the two primary

components with biggest variations are preserved. Here, the color of each pixel corresponds with the corresponding soft gating factor $W$, and a higher heat value of $W$ means a bigger compensation applied to transfer the feature embedding. We can see that bigger compensations are mostly applied to the data points locating in a self-contained region. This coincides with the principle that transfer learning is in general focused on such data examples that are incompatible with the target domain. Figure 2(b) shows the 2-dimensional histogram of maximum cosine similarity against soft gating factor $W$, which shows such a trend that lower similarity corresponds with in general higher compensation level, say, higher soft gating. Since the maximum similarity to cluster centers indicates the membership degree of the data example in association with a deterministic cluster, and a smaller value means a higher ambiguous case in terms of membership association, it is thus natural to result in higher-level transferring for feature compensation. This fully exhibits the motivation of this study, that is, why we make use of clustering centers as the anchors to figure out the contextual clue in directing feature compensation.

## 4.9 NETWORK DETAILS

Either MLP in Figure 1 contains a two-layer network. Table 5 shows the details. As for the Auto-Encoder (DEC) in Figure 1, Encoder/Decoder is composed of 4 linear layers, and the number of the neurons of each layer for the encoder is 1000,1000, 4000, and 100, respectively. Details of Auto-Encoder are listed in Table 6.

Table 5: Network Details of MLP (K is the number of cluster centers and 2048 is the feature dimension of ResNet)

|  | $\Delta$= MLP(X©S) | | W=MLP(S) | |
| --- | --- | --- | --- | --- |
|  | Layer 1 | Layer 2 | Layer 1 | Layer 2 |
| #Input | 2048+K | 2×(2048+K) | K | 2×K |
| #Neurons | 2×(2048+K) | 2048 | 2×K | 1 |
| #Output | 2×(2048+K) | 2048 | 2×K | 1 |
| Activation | GELU | - | GELU | Sigmoid |

Table 6: DEC Auto-Encoder Details (2048 is the feature dimension of ResNet)

|  | Encoder | | | Decoder | | |
| --- | --- | --- | --- | --- | --- | --- |
|  | #Input | #Neurons | #Output | #Input | #Neurons | #Output |
| Layer 1 | 2048 | 1000 | 1000 | 100 | 4000 | 4000 |
| Layer 2 | 1000 | 1000 | 1000 | 4000 | 1000 | 1000 |
| Layer 3 | 1000 | 4000 | 4000 | 1000 | 1000 | 1000 |
| Layer 4 | 4000 | 100 | 100 | 1000 | 2048 | 2048 |

## 5 CONCLUSION

We propose a new transfer learning method with context-aware compensation to feature embedding, where a contextual descriptor figuring out how close from the input to each cluster center is applied, and soft gating is used to make the feature transfer focused on the data samples that truly need transferring. The experiments demonstrate that the proposed feature transferring scheme is technically sound, and a big performance improvement is achieved using a tiny network. The strong point aspect as well as the weakness lies in the clustering-based contextual description, which determines the overall performance. In the future, we will investigate into more contextual description schemes.

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
