# OpenReview forum: "Transfer Learning with Context-aware Feature Compensation"
_ICLR.cc/2023/Conference — Submitted to ICLR 2023_

### Official Review · Reviewer_thJK · 2022-10-17

**Confidence:** 4
**Correctness:** 3
**Technical Novelty And Significance:** 1
**Empirical Novelty And Significance:** 3
**Recommendation:** 3

**Clarity, Quality, Novelty And Reproducibility:**

The paper is clear, as stated before. I also appreciate the fact that the authors have attached the project source code, allowing for reproducibility. However, the methodology is not particularly sound, neither technically nor empirically.

**Strength And Weaknesses:**

Strengths:
- The problem addressed by the paper is relevant.
- The methodology is clear and well-described.
- The experiments are fair and show that the method is competitive with SOTA in some cases.

Weaknesses:
1. In the introduction, the authors remark that "due to the overlap between the source domain and the target domain, not every data example needs transferring, and how much transferring is needed for a data example is subject to its position in the overall data distribution". Nonetheless, in their algorithm, clustering is performed using the target data. Thus, the computed similarities will not tell how far a given target example is from the source distribution. Therefore, I don't see how they are useful to assess "how much transferring is needed" for that example. When the source distribution is very different from the target, the method seems inappropriate. The poor results on MIT Indoor-67 reinforce this suspicion.
2. Are the displacements $\Delta$ normalized? Otherwise, the role of the predicted magnitude $W$ is redundant.
3. The experimental results are not particularly convincing. In particular, it is unclear which of the training strategies proposed in section 3.3 is the best one. In addition, it would be interesting to visualize what kind of transformations the model is learning, i.e. if it's learning $\Delta$s that pull representations towards the cluster centroids (or something else).
4. A theoretical analysis of the proposed methodology is missing. It is presented as a mere heuristic without any theoretical guarantees.

**Summary Of The Paper:**

The paper proposes an end-to-end architecture from transfer learning that learns a feature compensation for the input data. For this purpose, a clustering of the data is performed on a pre-processing step and the similarities between an input example and each of the clusters are used as inputs to two MLPs that essentially decide i) the magnitude ($W$) of the transformation to apply to the given input example and ii) the direction ($\Delta$) of that transformation. The method is validated on a set of benchmark datasets for transfer learning from ImageNet, using different backbone pre-trained networks. Different training strategies are compared.

**Summary Of The Review:**

The paper lacks any theoretical guarantees. Moreover, as explained in 1. in the section "Weaknesses", I am not fully convinced by the appropriateness of the proposed method. The experimental results are not excellent either. All these issues justify my recommendation for rejection.

---

### Official Review · Reviewer_9GK7 · 2022-10-24

**Confidence:** 3
**Correctness:** 2
**Technical Novelty And Significance:** 2
**Empirical Novelty And Significance:** 2
**Recommendation:** 3

**Clarity, Quality, Novelty And Reproducibility:**

Although the writing in general is clear, the proposed method is not clear and it is only very briefly explained in half a page. The most important technical decisions remain unexplained / unjustified. This also undermines the quality and novelty of the paper, which at this stage I cannot clearly see. The reproducibility of the paper is fair for researchers in the area.

**Strength And Weaknesses:**

Strengths:
- The paper tackles an important problem of identifying the mismatch between a source domain and a target domain before transfer.
- The paper is, in general, well-aligned with literature in the area.

Weaknesses:
- The paper does not provide explanations / intuitions to most technical decisions. For instance, the proposed method is very quickly explained in roughly half a page (sections 3.1 - 3.2). For example, why is it reasonable to use DEC? Why is compensation necessary - what are the underlying factors that make it necessary once the closest cluster centers have been identified? Why is cosine similarity, and no other measure, applicable in this case?
- It is very unclear to me how the proposed method is actually helping to identify the mismatch between a source domain and a target domain. In several parts of the paper, it is mentioned that the cluster centers are calculated on the full training set, and according to my understanding it is the whole training set of the target domain for which cluster centers are being calculated? I think it would greatly benefit the explanation of the method if it would clearly distinguish which parts of it come from the source domain and which from the target domain.
- In the experiments, I do not see any reasoning as to what the proposed method works in some cases but not in others. In which cases doing this kind of clustering + compensation approach would work better? This kind of description is very important to really see the contribution of the proposed method to the problem of transfer learning.

**Summary Of The Paper:**

This paper proposes a method for identifying the mismatch between a source domain and a target domain for transfer learning. The method relies on first identifying cluster centers using all the training data, and then comparing the current input data example to these center for determining how much compensation should be applied for the network to adapt to that particular example. This method requires to add two MLP network to the network architecture: one for compensation, and a second for soft gating this compensation. The experiments consider 5 benchmark datasets which are trained as downstream tasks of a backbone network trained on ImageNet-1k. The authors test several network architectures. The results show that the proposed method gains in terms of classification precision in some cases.

**Summary Of The Review:**

As explained in the 'Strengths and Weaknesses' section, I think that the paper lacks a lot of technical details regarding the proposed method and key technical decisions. The method itself does not seem to related well with the original research question (i.e. the mismatch between a source and a target domain). Finally, the authors do not provide insights into the experimental results that explain in which scenarios would the method be applicable, which undermines the contribution of the paper. For these reasons, I recommend a reject.

---

### Official Review · Reviewer_AuzT · 2022-10-24

**Confidence:** 5
**Correctness:** 3
**Technical Novelty And Significance:** 3
**Empirical Novelty And Significance:** Not applicable
**Recommendation:** 3

**Clarity, Quality, Novelty And Reproducibility:**

**Clarity:** The method part lacks clarity.

**Quality:** Medium quality

**Novelty:** Somewhat novel.

**Reproducibility:** Authors provides code, which seems reproducibility can be qualified.

**Strength And Weaknesses:**

### Strength

- The method is interesting. Studying the contextual relation between source and target domains is novel.
- The related work section is sufficient. So much literature is surveyed.
- Experimental datasets and comparison methods are representative.

### Weakness

- Although the idea of learning contextual relation is novel, this approach is not. It remains unclear whether clustering can model the context information, or if the results of clusters and original inputs can solve the mismatch between source and target domains in transfer learning.
- The claim that "this method only adds two MLPs to the backbone network" is not correct. As can be seen from Figure 1, there are more than two MLPs: it still has an autoencoder (DEC), which is much more complicated than MLPs.
- The method part is extremely confusing and not clear, which should be rewritten. I find figure 1 hard to understand, specifically how to compute $\mathbf{X} \textcopyright \mathbf{S}$ and $\mathbf{W} \cdot \Delta$. (I even think that should be $\cdot$ instead of $.$, since you claim it's a product.)
- The success of "compensation" heavily relies on DEC (deep embedding clustering). If DEC does not give the correct clustering results, the method will fail. How to justify this? Additionally, DEC brings extra computations, which should be highlighted in the paper.
- This method will consume a lot of training time than comparison methods, according to Sec. 3.3. We see that there is more than one step of fine-tuning, which is more complicated than existing work. Authors should compare your training and inference time in the experiments.
- In experiments, you should also list the number of training parameters of each method for a more fair study since this method introduces extra neural network modules.
- More importantly, the performance on general fine-tuning tasks of this method is NOT good. According to table 1, This approach only succeeds using Resnet-101, while it does not generate competitive results in Resnet-50 and VGG16. Hence, it is difficult to say that this approach actually works in different backbones.
- No experiments to explain why this method works. Lack of detailed analysis.
- With regards to related work, I think this paper misses two very important references on the same topic:

[1] Jang Y, Lee H, Hwang S J, et al. Learning what and where to transfer[C]//International Conference on Machine Learning. PMLR, 2019: 3030-3039.

[2] Murugesan K, Sadashivaiah V, Luss R, et al. Auto-Transfer: Learning to Route Transferrable Representations[J]. arXiv preprint arXiv:2202.01011, 2022.

**Summary Of The Paper:**

This paper studies how to effectively fine-tune a pre-trained network for a downstream task. The challenge here is to identify the mismatch between source and target distributions. To this end, this paper proposes context-aware feature compensation, which learns context information among the training data. They first perform clustering on the training data, which is used to learn cluster-to-data similarity matrix. Then, the similarity matrix is used to generate compensation embeddings using two MLPs. The compensation embeddings are then concatenated with the original training data. Experiments are performed on different fine-tuning datasets, with a broad comparison of recent fine-tuning methods.

**Summary Of The Review:**

This paper aims to tackle transfer learning in context view. I find it a little novel, but the approach is not novel, with so many flaws, as listed in weakness section. Experimental results are not comparable, with missing references. Therefore, I tend to reject this paper.

---

### Official Review · Reviewer_a21v · 2022-11-06

**Confidence:** 4
**Correctness:** 2
**Technical Novelty And Significance:** 2
**Empirical Novelty And Significance:** 2
**Recommendation:** 3

**Clarity, Quality, Novelty And Reproducibility:**

Clarity 3/4
Quality 2/4
Reproducibility 3/4
However, the paper lacks novelty, its technique lacks theories backing it up and not sound at all.

**Strength And Weaknesses:**

Strengths:
the paper is well written, methodology is well-described, and the problem addressed is relevant.

Weaknesses.
The author is describing how to use data centers to do data compensation, however, how this is related to data transferring is poorly described. For examples, is data centers computed on target data set? how is its feature vector computed, using the pretrained network? using distance to get compensation factor seems only to be working under a strong assumption that this is used for classification, and it only works for image classification tasks? why using distance to centers work? all these questions are not answered properly.

Based on the experimental setup, it is just finetuning the pretrained network using some downstream tasks. It just barely competes with SOTA methods, seems that with or without such 2 layers of compensation networks does not change the result at all.

**Summary Of The Paper:**

The paper proposes an feature compensation method by two MLPs that computes the distance of data points to data centers. .The method is evaluated on a set of benchmark datasets for transfer learning from ImageNet, showing somewhat on par result using this technique.

**Summary Of The Review:**

This paper is well written and it addresses a problem relevant to transfer learning. However, it is just an empirical work that lacks theory backing the idea, such that using distance to data clusters to do compensation as data transferring. Finally, the results are just on par with existing SOTA methods, not proving the effectiveness of this method.

---

### Decision · Program_Chairs · 2023-01-20

**Decision:**

Reject

**Justification For Why Not Higher Score:**

Motivation and detailed explanations of the proposed method are missing. Experimental results are not convincing.

**Justification For Why Not Lower Score:**

N/A

**Metareview: Summary, Strengths And Weaknesses:**

In this paper, the authors proposed a feature compensation-based transfer learning method.

Though the high-level idea is interesting, the specific design of the proposed method is technically incremental. In addition, the motivation for the proposed method is not clear. This is a lack of explanations on the technical components of the proposed method. Experimental results fail to verify the effectiveness of the proposed method.

In summary, this paper is not ready for publication based on its current shape.